

# Contrasting effects of increasing dissolved iron on photosynthesis and O$_2$ availability in the gastric cavity of two Mediterranean corals

Walter Dellisanti[1], Qingfeng Zhang[1], Christine Ferrier-Pagès[2] and Michael Kühl[1]

[1] Department of Biology, Marine Biology Section, University of Copenhagen, Helsingør, Denmark
[2] Coral Ecophysiology Laboratory, Center Scientifique de Monaco, Principality of Monaco, Monaco

## ABSTRACT

Iron (Fe) plays a fundamental role in coral symbiosis, supporting photosynthesis, respiration, and many important enzymatic reactions. However, the extent to which corals are limited by Fe and their metabolic responses to inorganic Fe enrichment remains to be understood. We used respirometry, variable chlorophyll fluorescence, and O$_2$ microsensors to investigate the impact of increasing Fe(III) concentrations (20, 50, and 100 nM) on the photosynthetic capacity of two Mediterranean coral species, *Cladocora caespitosa* and *Oculina patagonica*. While the bioavailability of inorganic Fe can rapidly decrease, we nevertheless observed significant physiological effects at all Fe concentrations. In *C. caespitosa*, exposure to 50 nM Fe(III) increased rates of respiration and photosynthesis, while the relative electron transport rate (rETR(II)) decreased at higher Fe(III) exposure (100 nM). In contrast, *O. patagonica* reduced respiration, photosynthesis rates, and maximum PSII quantum yield (F$_v$/F$_m$) across all iron enrichments. Both corals exhibited increased hypoxia (<50 $\mu$mol O$_2$ L$^{-1}$) within their gastric cavity at night when exposed to 50 and 100 nM Fe(III), leading to increased polyp contraction time and reduced O$_2$ exchange with the surrounding water. Our results indicate that *C. caespitosa*, but not *O. patagonica*, might be limited in Fe for achieving maximal photosynthetic efficiency. Understanding the multifaceted role of iron in corals' health and their response to environmental change is crucial for effective coral conservation.

# INTRODUCTION

Scleractinian corals are the foundation species of tropical coral reefs and temperate coralligenous assemblages (*Hoeksema, 2017*; *Ingrosso et al., 2018*). Their metabolic processes (*i.e.,* cellular respiration, symbiont photosynthesis, calcification), as well as cellular homeostasis (*Reich et al., 2023*), are linked to macro- and micro-nutrient availability through host feeding and nutrient uptake by endosymbionts (*Muscatine & Porter, 1977*; *Houlbrèque & Ferrier-Pagès, 2009*; *Houk et al., 2020*). Among micro-nutrients, iron is fundamental for cellular function and metabolism (*Raven, Evans & Korb, 1999*;

Corresponding author
Walter Dellisanti,
walter.dellisanti@gmail.com

*Behrenfeld & Milligan, 2013*; *D'Angelo & Wiedenmann, 2014*; *Reich et al., 2020*). However, the impacts of iron on coral metabolism are multifaceted.

Iron exists in seawater in two oxidation states, Fe(II) and Fe(III), and can be found as "free" ions or complexed with organic and inorganic ligands (*Liu & Millero, 2002*; *Blain & Tagliabue, 2016*). The availability of iron in aerated seawater is primarily as Fe(III), due to the complexation of Fe(II) by strong iron-binding ligands (*Johnson, Gordon & Coale, 1997*), such as inorganic oxides, extracellular polymeric exudates from phytoplankton, or siderophores produced by bacteria (*Walte & Morel, 1984*; *Gledhill et al., 2004*; *Hassler et al., 2011*). Dissolved iron is mostly adsorbed by photosynthetic symbionts and supports the photosynthetic process in corals *via* its role as a cofactor in proteins involved in photosynthesis, electron transport, and scavenging enzymes for reactive oxygen species (*Raven, Evans & Korb, 1999*; *Reichelt-Brushett & McOrist, 2003*; *Reich et al., 2020*). Iron deficiency can thus significantly limit coral photosynthesis, potentially resulting in reduced growth rates, compromised skeletal development, and impaired energy acquisition (*Entsch, Sim & Hatcher, 1983*; *Ferrier-Pagès et al., 2001*; *Shick et al., 2011*). However, exposure to high metal concentrations, including iron, can also disrupt the coral symbiosis and lead to bleaching (*Harland & Brown, 1989*; *Reichelt-Brushett & McOrist, 2003*; *Leigh-Smith, Reichelt-Brushett & Rose, 2018*). The availability of inorganic iron also influences the biogeochemical cycling of other trace metals (*Rodriguez et al., 2016*), acting as essential cofactors in numerous enzymatic and biochemical reactions (*Morrissey et al., 2015*; *Sutak, Camadro & Lesuisse, 2020*). Therefore, a disruption in trace metal dynamics may further exacerbate the physiological stress experienced by corals due to combined environmental changes (*i.e.,* rising temperatures, ocean acidification, nutrient pollution) (*Ponti et al., 2021*), leading to oxidative stress (*Leigh-Smith, Reichelt-Brushett & Rose, 2018*). This can ultimately impact the capacity of corals to withstand environmental changes, leading to decreased resilience and survival.

Key metabolic activities in corals occur in their gastric cavity (coelenteron), which plays an important role in processes like digestion, excretion, and internal nutrient cycling but remains largely unexplored (*Hughes et al., 2022*). Due to the presence of photosynthetic endosymbionts living in symbiosis within the coral endoderm facing the coelenteron, $O_2$ levels within the gastric cavity are strongly influenced by light conditions and can vary with depth. In the upper region of the gastric cavity, the $O_2$ levels can be as high as 400% air saturation (*Agostini et al., 2012*), while deeper regions of the gastric cavity can exhibit lower $O_2$ levels reaching hypoxia or even anoxia (*Agostini et al., 2012*), especially in darkness. Such $O_2$ dynamics influence the types of microorganisms that thrive within the gastric cavity (*Agostini et al., 2012*; *La Rivière, Garel & Bally, 2016*) and might also affect trace metal and nutrient availability *via* $O_2$-dependent redox changes in the gastric cavity. There is thus a need for further research to explore the role of trace metals, specifically dissolved iron, within the gastric cavity and to examine how iron enrichment might impact the availability of $O_2$ in this critical microenvironment.

While the effects of metals on coral physiology, have been studied in tropical corals (*Ferrier-Pagès et al., 2001*; *Reichelt-Brushett & McOrist, 2003*; *Reich et al., 2023*), this is not the case for temperate corals. These corals are often more exposed to anthropogenic

activities, which increase metal levels in seawater and sediments (*Mahowald et al., 2005*; *Krishnamurthy et al., 2010*). For example, Mediterranean corals are the key species of the marine animal forests (MAFs), which exhibit a high biodiversity of marine species and provide important ecosystem services and functions (*Fine, Zibrowius & Loya, 2001*; *Kružić & Benković, 2008*; *Ingrosso et al., 2018*; *Bevilacqua et al., 2021*). These forests receive diverse inputs of inorganic iron from both natural and anthropogenic sources, including atmospheric deposition, terrestrial runoff, and hydrothermal vents (*Sarthou & Jeandel, 2001*; *Bonnet & Guieu, 2006*; *Molari et al., 2018*). Such diverse sources contribute to spatial and temporal heterogeneity in iron concentrations and thus the availability of this trace metal to marine organisms (*Guerzoni et al., 1999*; *Wagener, Guieu & Leblond, 2010*; *Gallisai et al., 2014*). Iron levels in Mediterranean seawater typically vary from 0.7 to 14.5 nM in coastal regions (*Sarthou & Jeandel, 2001*), and iron enrichment has been found prevalent in coastal sediments near harbors in the Gulf of Genoa and the Adriatic Sea (*Bertolotto et al., 2005*; *Orlov et al., 2022*). In comparison, these iron levels are generally higher than those recorded in Caribbean and Indo-Pacific waters (<5 nM, *GEOTRACES Intermediate Data Product Group, 2021*). Within the Mediterranean Sea, *C. caespitosa* and *O. patagonica* are two key coral species serving as bioindicators for water quality in the Mediterranean Sea (*Fine, Zibrowius & Loya, 2001*; *Peirano et al., 2004*; *Rodolfo-Metalpa et al., 2006*; *Kružić & Benković, 2008*; *Casado de Amezua et al., 2015*) and they are naturally exposed to varying levels of inorganic trace metals. However, the role and effect of dissolved iron in these symbiont-bearing corals remains unknown.

In this study, we explore the effect of increasing Fe(III) levels on respiration and photosynthesis, as well as on the internal $O_2$ status of the gastric cavity, in two common Mediterranean corals, *Cladocora caespitosa* and *Oculina patagonica*. We hypothesized that low to moderate Fe(III) exposure may support their photosynthetic capacity and physiological status, while excessive amounts of Fe(III) above a certain threshold have detrimental effects on coral physiology. Our results indicate that a short-term enrichment of dissolved iron might be beneficial in the host-symbiont relationship of corals living in an iron-limited environment, providing new insights into the metabolic responses of corals to trace metals.

## MATERIALS & METHODS

Three healthy colonies (20 to 30 $cm^2$) of *Cladocora caespitosa* were collected from Trieste (Italy, 45.707°N, 13.712°E) using SCUBA in April 2023 (CITES IT/EX/2023/MCE/00335). Subsequently, the corals were carefully placed in a container with aerated seawater (50 L) and transported to the laboratory in Monaco. Ten colonies of *Oculina patagonica* (10 to 20 $cm^2$) were originally sampled in Albissola, Gulf of Genoa (Italy, 44.283°N, 8.50°E) as previously described (*Rodolfo-Metalpa et al., 2006*) and were kept in flow-through aquaria in Monaco that were continuously supplied with non-filtered seawater sampled from ~50 m depth and heated to 18 °C. Three colonies of *O. patagonica* were randomly selected and used in this study. Corals were exposed to a 12:12 h light-dark cycle under a photon irradiance (400–700 nm) of 150 ± 10 µmol photons $m^{-2}$ $s^{-1}$ provided by 400 W metal

**Figure 1** **Schematic representation of internal features of coral polyps.** (A) *Cladocora caespitosa*, and (B) *Oculina patagonica:* polyps size, gastric cavity, and skeleton.

halide lamps (HQI-TS, Philips). The coral colonies were fragmented into 48 nubbins ($n = 24$ per species) and placed on PVC support using epoxy resin putty. The nubbins of *C. caespitosa* consisted of single polyps $\sim$2 cm$^2$, and those of *O. patagonica* consisted of multiple polyps with an approximate size of $\sim$5 cm$^2$ (Fig. 1). Corals were fed twice a week with *Artemia salina* nauplii and were allowed to acclimate to the aquarium conditions for two weeks before the onset of experiments.

The experimental design consisted of four 8 L tanks; one tank was maintained without Fe(III) addition, while the other tanks were used for the iron enrichment experiments. The tanks were filled with the same non-filtered seawater and maintained under controlled conditions in a water bath at 18 °C with a photoperiod of 12:12 and an incident photon irradiance (400–700 nm) of $150 \pm 10$ µmol photons m$^{-2}$ s$^{-1}$, as measured with a Universal Light Meter (ULM-500) equipped with a spherical micro quantum sensor (US-SQS/L, Heinz Walz, Effeltrich, Germany). Following two weeks of acclimation, the nubbins ($n = 6$ per species per condition) were randomly mixed and transferred into the tanks where they were kept unfed.

A stock solution of Fe(III) (50 µM FeCl$_3$*6H$_2$O, Sigma-Aldrich, St. Louis, MO, USA) was prepared for the iron enrichment experiments. Two pulses of Fe(III) were added per day by diluting the stock solution to expose corals to 0, 20, 50, and 100 nM Fe/day, hereafter named "control", "20Fe", "50Fe", and "100Fe", and the experiment lasted one week. No water renewal was carried out during the experimental period. After one week of exposure, seawater samples (15 mL) were collected in metal-free tubes (Labcon) from the tanks for elemental iron determination and from the respirometry chambers for measuring iron uptake (see below). The final iron concentration was measured in 0.32 M HNO$_3$ diluted samples using inductively coupled plasma mass spectrometry (ICP-MS, iCAP-Q, Thermo Scientific, Waltham, MA, USA). The uptake rate was calculated as the absolute difference of Fe measured in the chambers from the Fe measured in the tanks and Fe concentration was expressed in µg L$^{-1}$.

The net production/consumption of dissolved oxygen (O$_2$) due to coral photosynthesis and respiration was measured using custom-made respirometry chambers (55 mL) composed of transparent polycarbonate (Fig. S1). The chambers were closed with a gas-tight transparent lid and contained a magnetic stirrer below a perforated plate at the

bottom to ensure water mixing during incubation. An $O_2$-sensitive optical sensor spot (OXSP5-ADH, Pyroscience GmbH, Aachen, Germany) was attached to each chamber's internal surface and was read out *via* a fiber optic cable (SPFIB-LNS, Pyroscience GmbH, Aachen, Germany) fixed *via* a holder on the outside of the transparent chamber wall and connected to a fiber-optic $O_2$ meter (FSPRO-4, Pyroscience GmbH, Aachen, Germany). The meter was connected *via* a USB cable to a PC running the data-logging software (Pyro Workbench; Pyroscience GmbH). Prior to experimental measurements, sensors were calibrated in $\mu mol\ O_2 L^{-1}$ using a 2-point calibration, measuring the sensor signals in anoxic (seawater with $Na_2SO_3$) and 100% air-saturated seawater at experimental temperature and salinity.

Dissolved oxygen was recorded every second during the incubation of corals. Dark respiration (R) and net photosynthesis ($P_n$) were calculated from the linear change in $O_2$ concentration over time measured during 30-min dark and 30-min light incubations, and the rates were calculated as $(\Delta O_2/\Delta t) \times V/A$, where V is the volume of seawater surrounding the coral samples in the chamber and A is the coral surface area. The coral surface area was determined with the aluminum foil technique (*Marsh, 1970*). The surface area of the aluminum foil used to cover living tissue was determined using the ImageJ software v.1.53 (*Schneider, Rasband & Eliceiri, 2012*).

Gross photosynthesis ($P_g$) was calculated by adding the absolute value of R to $P_n$, assuming that the dark respiration was identical to respiration in the light. Subsequently, $P_g$:R ratios were calculated as a measure of the diurnal productivity and degree of autotrophy of the coral holobiont. We note that while $P_g$ can be assigned to the photosynthesis of endosymbionts, R is affected by the respiratory activity of the coral host, its photosynthetic endosymbionts as well as its microbiome.

The photosynthetic capacity of the endosymbionts in the coral samples was assessed with variable chlorophyll fluorimetry using a Pulse Amplitude Modulated (PAM) fluorometer (Dual-PAM, Heinz Walz, Effeltrich, Germany) equipped with a standard glass-fiber optic probe (*Ralph et al., 1999*). These measurements were obtained at the end of each iron enrichment period from single polyp measurements of each coral ($n = 5$ per condition), after 15 min dark acclimation. The maximum photochemical quantum yield of PSII ($F_v/F_m$) was calculated as (*Schreiber, 2004*): $F_v/F_m = (F_m - F_0)/F_m$, where $F_m$ is the maximum fluorescence yield measured during a strong saturation pulse (3,000 $\mu$mol photons $m^{-2}\ s^{-1}$, width 600 ms) and $F_0$ isthe minimum fluorescence yield before the saturation pulse using weak measuring light pulses (<1 $\mu$mol photons $m^{-2}\ s^{-1}$, width 3 $\mu$s, frequency 0.6 kHz). The minimum fluorescent yield, $F_0$, is measured when all PSII centers are open and can be used as a proxy for chlorophyll biomass (*Serôdio, Da Silva & Catarino, 1997*), while the maximum fluorescence yield, $F_m$, is measured when all PSII centers are closed in response of the saturation pulse (*Baker et al., 2001*). Rapid light curves (RLCs) were measured by illuminating dark-adapted corals at increasing irradiance from 0 to 2,000 $\mu$mol photons $m^{-2}\ s^{-1}$ (PAR) with 20 s incubation at each irradiance step (*Ralph & Gademann, 2005*; *Trampe et al., 2011*). The effective photochemical quantum yield of PS(II) was calculated as $Y(II) = (F'_m - F)/F_m$ (*Genty, Briantais & Baker, 1989*) and provides a measure of the PSII photosynthetic capacity. The rETR was calculated from $Y(II)$ and

the actinic photon irradiance, $E_d$, as rETR $=$ Y(II) $\times$ $E_d$ and represents a relative measure of the PSII electron transport rate (*Beer et al., 1998*). All rETR, $F_v/F_m$, and Y(II) yields were calculated using the system software (WinControl, Heinz Walz GmbH, Effeltrich, Germany).

Clark-type, electrochemical $O_2$ microsensors with a slender shaft and a tip diameter of 50 $\mu$m (OX-50, Unisense A/S, Denmark) were used to measure the $O_2$ distribution within the gastric cavity of corals under light exposure and in darkness. Additionally, overnight $O_2$ fluctuations (6 p.m. to 6 a.m.) were continuously recorded in the dark with the microsensor tip positioned in the lower part of the gastric cavity.

For microsensor measurements, coral polyps ($n = 3$) were placed inside a custom-designed flow chamber (0.8 L), with a consistent laminar water flow (0.5 cm s$^{-1}$) of oxygenated seawater (18 °C and salinity of 35) as previously described by *Haro et al. (2019)*. During light incubation, the corals were illuminated with a fiber optic lamp (KL 2500 LED, Schott) with known photon irradiance (400–700 nm) levels of 150 $\pm$ 10 $\mu$mol photons m$^{-2}$s$^{-1}$, as determined by a Universal Light Meter (ULM-500) equipped with a spherical micro quantum sensor (US-SQS/L, Heinz Walz, Effeltrich, Germany). The $O_2$ microsensors were linearly calibrated from sensor signal readings in 100% air-saturated seawater and anoxic water (using sodium ascorbate solution). The $O_2$ microsensor was mounted on a motorized micromanipulator system (Unisense A/S, Denmark) and was connected to a microsensor meter (fx-6 UniAmp, Unisense A/S, Denmark). The micromanipulator and microsensor meter were connected to a PC running dedicated software for data acquisition and sensor positioning (SensorSuite Profiler v3.2, Unisense A/S, Denmark). The positioning of the microsensor tip relative to the coral surface was observed *via* a dissection microscope and a digital USB microscope (Dino-Lite 5MP Edge, AnMo Electronics, Taiwan).

For measurements in the gastric cavity, one depth profile was measured per coral polyp. The microsensor tip was initially positioned at the center of the coral mouth, after which was lowered vertically using the micromanipulator by 50–100 $\mu$m steps until a contraction of the coral polyp was observed. This position was identified as the base of the gastric cavity. After the microsensor signal was stable (minimum 10 min), depth profiles of $O_2$ concentration were measured by moving stepwise from the cavity's base toward the mouth and concluding ($\sim$2 mm) in the seawater above the coral mouth. For tracking temporal $O_2$ variations overnight within the gastric cavity under dark conditions, the microsensor remained positioned at the base of the coral gastric cavity. One temporal profile was measured per coral polyp. Concurrently, a digital microscope (Dino-Lite 5MP Edge, AnMo Electronics, Taiwan) was employed to capture time-lapse imagery of coral contractions simultaneously with the $O_2$ concentration measurements (https://doi.org/10.5281/zenodo.10698045), with frames captured every 2 min using DinoCapture 2.0 software. The time-lapse videos were checked for polyp contraction every three frames (every 6 min in real-time), and the contraction was marked by tentacle retraction and reduction in the relative height of the oral disk. The percentage of contraction time was calculated by dividing the number of frames in which contraction occurs by the total number of examined frames.

All data were log10-transformed and checked for normality using the Shapiro–Wilk test and for homogeneity of variance using Levene's test. When data did not meet the

**Table 1  Experimental conditions and Fe uptake in *C. caespitosa* and *O. patagonica*.** Condition indicates the pulses of Fe(III) added to expose corals to 0, 20, 50, and 100 nM Fe/day. Fe(III) uptake is calculated as the difference between Fe concentrations in the seawater and in the chambers at the end of incubation.

| Species | Condition | Fe(III) seawater $\mu g\,L^{-1}$ | Fe(III) uptake $\mu g\,cm^{-2}\,h^{-1}$ | Chi-squared | Df | p-value |
|---|---|---|---|---|---|---|
| *C. caespitosa* | Control | $1.67 \pm 0.12$ | $0.01 \pm 0.07$ | 9.154 | 3 | <0.05 |
| | 20 nM | $1.77 \pm 0.15$ | $0.20 \pm 0.15$ | | | |
| | 50 nM | $1.50 \pm 1.53$ | $0.33 \pm 0.16$ | | | |
| | 100 nM | $2.29 \pm 0.12$ | $0.01 \pm 0.03$ | | | |
| *O. patagonica* | Control | $1.67 \pm 0.12$ | $0.00 \pm 0.03$ | 4.378 | 3 | *n.s.* |
| | 20 nM | $1.77 \pm 0.15$ | $0.09 \pm 0.14$ | | | |
| | 50 nM | $1.50 \pm 1.53$ | $0.07 \pm 0.33$ | | | |
| | 100 nM | $2.29 \pm 0.12$ | $0.00 \pm 0.04$ | | | |

Notes.
Kruskal–Wallis test, Chi-squared, test statistic H; Df, degrees of freedom.
Values are average and standard deviation.

assumptions of normality, a Kruskal–Wallis test was used with the Wilcoxon rank test for pairwise comparison. Parametric one-way analysis of variance (ANOVA) with the Tukey HSD test was used for data that followed a normal distribution. A linear regression model was used to estimate the oxygen variability considering contraction time, condition, and species as predictor variables. Finally, a multivariate analysis of variance (MANOVA) was used to verify significant differences in iron enrichment between species and condition groups and visualized with a Principal Component Analysis (PCA). All statistical analyses were run in R v4.2.3 using *dplyr* (*Wickham et al., 2023*) and *multcomp* packages (*Hothorn, Bretz & Westfall, 2008*) and visualized with the *ggplot2* package (*Wickham, 2016*).

# RESULTS

The levels of dissolved iron measured in tanks ranged between 1.5 to $2.29 \pm 0.12\ \mu g\,L^{-1}$ (Table 1). *Cladocora caespitosa* exhibited a significant iron uptake with the highest uptake rate reaching $0.33 \pm 0.16\ \mu g\,cm^{-2}\,h^{-1}$ in the 50 nM Fe(III) treatment, and the lowest in the 100 nM Fe(III) treatment ($0.01 \pm 0.03\ \mu g\,cm^{-2}\,h^{-1}$) (Kruskal–Wallis test, $X^2 = 9.154$, $df = 3$, $p < 0.05$; Table 1). When comparing the individual conditions (Wilcoxon rank test) within *C. caespitosa* and *O. patagonica* groups, no significant differences were detected.

## Coral metabolic rates

Iron uptake had divergent effects on the physiological characteristics of the two coral species (Fig. 2). Specifically, in *C. caespitosa*, the respiration rate increased from $0.16 \pm 0.02\ \mu mol$ $O_2\,cm^{-2}\,h^{-1}$ in the 50 nM Fe(III) treatment to $0.29 \pm 0.08\ \mu mol\,O_2\,cm^{-2}\,h^{-1}$ in the 100 nM Fe(III) treatment (Kruskal–Wallis test, $X^2 = 11.6$, $df = 3$, $p < 0.01$). A similar pattern was observed in photosynthetic rates, where net photosynthesis increased from $0.14 \pm 0.02\ \mu mol\,O_2\,cm^{-2}\,h^{-1}$ in the 20 nM Fe(III) to $0.29 \pm 0.07\ \mu mol\,O_2\,cm^{-2}$ $h^{-1}$ in 50 and 100 nM Fe(III) (Kruskal–Wallis test, $X^2 = 12.2$, $df = 3$, $p < 0.01$), and gross photosynthesis increased from $0.3 \pm 0.04\ \mu mol\,O_2\,cm^{-2}\,h^{-1}$ in the 20 nM Fe(III)

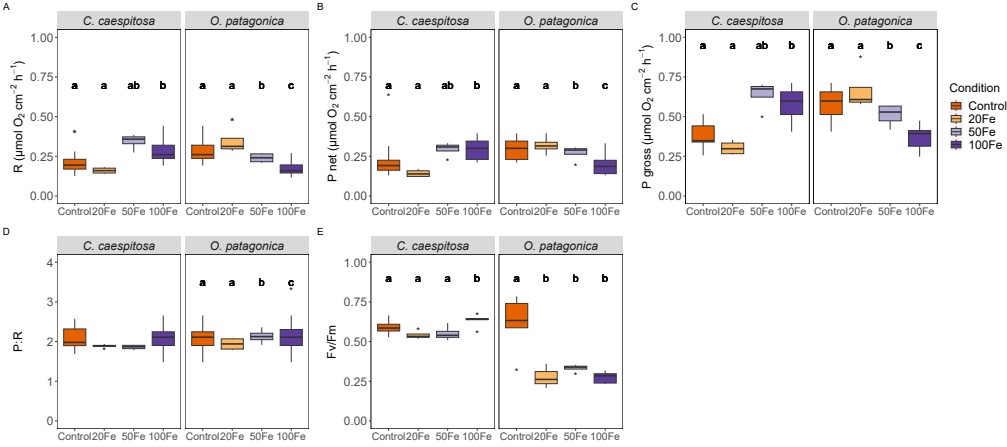

**Figure 2** **Physiological parameters of corals under increasing Fe(III) exposure.** (A) Respiration rate, (B) net photosynthesis rate, (C) gross photosynthesis rate, (D) ratio of gross photosynthesis to respiration (P:R), and (E) maximum quantum efficiency of Photosystem II ($F_v/F_m$) in *C. caespitosa* and *O. patagonica*. Condition indicates the pulses of Fe(III) added to expose corals to 0, 20, 50, and 100 nM Fe/day. The letters above the bars indicate significant differences among conditions ($p < 0.05$).

treatment to $0.64 \pm 0.09$ μmol $O_2$ cm$^{-2}$ h$^{-1}$ in the 50 nM Fe(III) and 100 nM Fe(III) treatments (Kruskal–Wallis test, $X^2 = 11.8$, $df = 3$, $p < 0.01$). No detectable effect of iron enrichment on the P:R ratio of *C. caespitosa*, but the maximum quantum yield of PSII ($F_v/F_m$) increased slightly from $0.54 \pm 0.03$ in the 20 nM Fe(III) treatment to $0.63 \pm 0.04$ in the 100 nM Fe(III) treatment (ANOVA, $F = 3.83$, $p < 0.05$, Table S1).

On the other hand, *O. patagonica* exhibited a decrease in respiration rates, continuously dropping from $0.35 \pm 0.09$ at 20 nM Fe(III) to $0.18 \pm 0.05$ μmol $O_2$ cm$^{-2}$ h$^{-1}$ in the 100 nM Fe(III) treatment (Kruskal–Wallis test, $X^2 = 19.1$, $df = 3$, $p < 0.01$). A similar pattern was observed in photosynthetic rates, where net photosynthesis decreased at Fe(III) levels >20 nM Fe(III) from $0.32 \pm 0.06$ to $0.20 \pm 0.07$ μmol $O_2$ cm$^{-2}$ h$^{-1}$ (Kruskal–Wallis test, $X^2 = 8.54$, $df = 3$, $p < 0.05$), and gross photosynthesis continuously decreased at Fe(III) levels >20 nM Fe(III), from $0.67 \pm 0.14$ to $0.37 \pm 0.07$ μmol $O_2$ cm$^{-2}$ h$^{-1}$ (Kruskal–Wallis test, $X^2 = 19.4$, $df = 3$, $p < 0.01$). A slight increase was observed in the P:R ratio of *O. patagonica*, from $1.94 \pm 0.16$ in the 20 nM Fe(III) treatment to $2.13 \pm 0.18$ and $2.21 \pm 0.57$ in the 50 nM and 100 nM Fe(III) treatments, respectively (ANOVA, $F = 8.29$, $p < 0.01$). The $F_v/F_m$ significantly decreased in *O. patagonica* from $0.61 \pm 0.18$ in the control condition to $0.28 \pm 0.08$ at >20 nM Fe(III) (ANOVA, $F = 10.94$, $p < 0.01$, Table S1).

### Variable chlorophyll fluorescence measurements

Differences in variable chlorophyll fluorescence parameters were evident between the two coral species across iron enrichment levels, particularly in terms of the minimum and maximum fluorescence yield, the PSII quantum yield, Y(II), and the derived relative electron transport rate, rETR, during rapid light curve measurements (Fig. 3, Table S2). In *C. caespitosa*, the exposure to 20 nM Fe(III) resulted in a significant enhancement of the minimum (Kruskal–Wallis test, $X^2 = 92.187$, $df = 3$), and maximum fluorescence yield

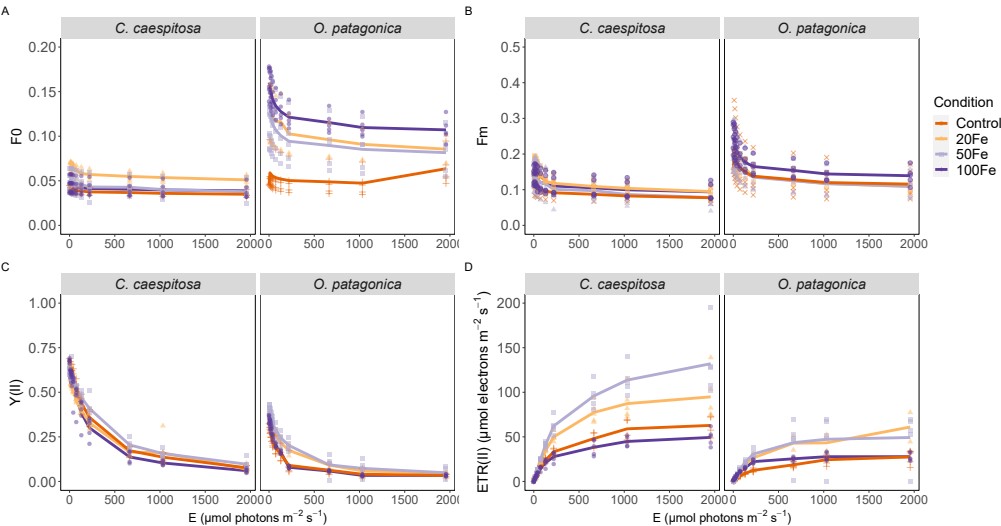

**Figure 3** **Variable chlorophyll fluorescence measurements.** (A) Minimum fluorescence yield; (B) maximum fluorescence yield; (C) effective PSII quantum yield, YII, and (D) derived relative electron transport rates, rETR, as a function of rapidly increasing photon irradiance, *i.e.,* RLC measurements. Condition indicates the pulses of Fe(III) added to expose corals to 0, 20, 50, and 100 nM Fe/day. The continuous line indicates the mean values.

(Kruskal–Wallis test, $X^2 = 20.085$, $df = 3$, $p < 0.01$). The effective quantum yield, Y(II), remained unaffected, while the relative electron transport rate, rETR, showed a decrease in the 100 nM Fe(III) treatment (Kruskal–Wallis test, $X^2 = 9.225$, $df = 3$, $p < 0.05$).

In *O. patagonica*, the exposure to 100 nM Fe(III) resulted in a significant enhancement of minimum (Kruskal–Wallis test, $X^2 = 110.06$, $df = 3$) and maximum fluorescence yields (Kruskal–Wallis test, $X^2 = 17.625$, $df = 3$, $p < 0.01$). A higher effective quantum yield, Y(II), was observed at 20 nM and 50 nM Fe(III) treatment (Kruskal–Wallis test, $X^2 = 15.109$, $df = 3$, $p < 0.01$), while no significant variations were observed in the relative electron transport rate.

## Spatial and temporal O$_2$ profiles

The O$_2$ availability within the gastric cavity of *C. caespitosa* and *O. patagonica* exhibited significant differences across a depth range of 1–2 mm from the polyp's surface (Fig. 4A, Table S3). Specifically, the availability of O$_2$ in *C. caespitosa* under dark conditions decreased when transitioning from control conditions to iron enrichment conditions (Kruskal–Wallis test, $X^2 = 1.5$, $df = 3$, $p < 0.01$). Notably, the exposure to 50 nM and 100 nM Fe(III) resulted in hypoxic to anoxic conditions (<50 $\mu$mol O$_2$) in the dark. In the presence of 100 nM Fe(III), we also found a marked reduction in O$_2$ availability when corals were exposed to light (Kruskal–Wallis test, $X^2 = 32.9$, $df = 3$, $p < 0.01$). A similar trend emerged in the gastric cavity of *O. patagonica* when subjected to increasing Fe(III) levels, which led to diminished O$_2$ availability in comparison to the control under both dark (Kruskal–Wallis test, $X^2 = 1.3$, $df = 3$, $p < 0.01$) and light (Kruskal–Wallis test, $X^2 = 11.1$, $df = 3$, $p < 0.05$) conditions.

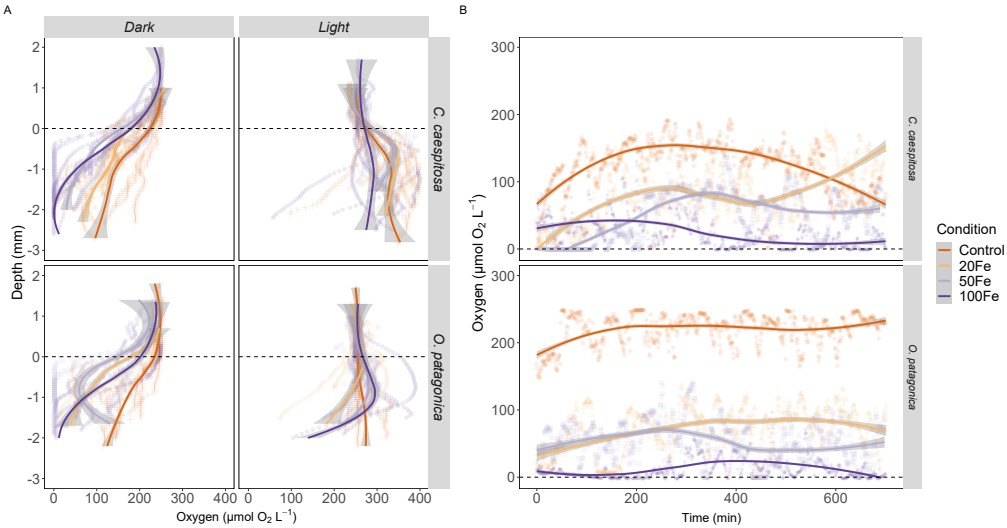

**Figure 4** **Oxygen distribution and dynamics in the gastric cavity of coral specimens.** (A) Depth profiles of $O_2$ concentration in the gastric cavity of *C. caespitosa* and *O. patagonica* measured under dark and light conditions; (B) temporal variation in $O_2$ concentration measured in *C. caespitosa* and *O. patagonica* under dark condition in the bottom of the gastric cavity overnight. Condition indicates the pulses of Fe(III) added to expose corals to 0, 20, 50, and 100 nM Fe/day. The dotted line indicates the polyp surface level. The continuous line indicates the trend line using *loess* method, with the shaded area as confidence intervals (95%).

The $O_2$ levels measured overnight within the gastric cavity of *C. caespitosa* and *O. patagonica* displayed significant variations based on different Fe levels (Fig. 4B). Specifically, when transitioning from control conditions to Fe(III) enrichment conditions, the availability of $O_2$ in *C. caespitosa* was significantly reduced from $125 \pm 34.3$ μmol $O_2$ $L^{-1}$ under control conditions to $80.2 \pm 43.2$ μmol $O_2$ $L^{-1}$ at 20 nM Fe(III), $48.3 \pm 36.3$ μmol $O_2$ $L^{-1}$ at 50 nM Fe(III), and $23.8 \pm 23.2$ μmol $O_2$ $L^{-1}$ at 100 nM Fe(III) (Kruskal–Wallis test, $X^2 = 14.5$, $df = 3$, $p < 0.01$). Similarly, in the case of *O. patagonica*, $O_2$ availability decreased in the presence of high Fe(III) levels, from $219 \pm 23$ μmol $O_2$ $L^{-1}$ in control conditions to $71.3 \pm 32.8$ μmol $O_2$ $L^{-1}$ at 20 nM Fe(III), $52.3 \pm 31.1$ μmol $O_2$ $L^{-1}$ at 50 nM Fe(III), and $11.7 \pm 15$ μmol $O_2$ $L^{-1}$ at 100 nM Fe(III) (Kruskal–Wallis test, $X^2 = 21.0$, $df = 3$, $p < 0.01$).

## Coral tissue contraction

The $O_2$ levels within the gastric cavity of both *C. caespitosa* and *O. patagonica* corals exhibit a significant correlation with the contraction time of the polyp ($p < 0.01$, Fig. 5, Table S4). The linear regression model results indicated a significant association between contraction time, Fe(III) exposure, and the $O_2$ level in the gastric cavity. Fe(III) enrichment had a notable effect on contraction time, as reflected by its negative coefficient estimate of $-29.80$ ($p < 0.01$). This negative coefficient suggests that, on average, when the corals are exposed to higher Fe(III) levels, the tissue tends to contract more. Consequently, an increase in contraction time is associated with a reduction in $O_2$ levels in both *C. caespitosa*
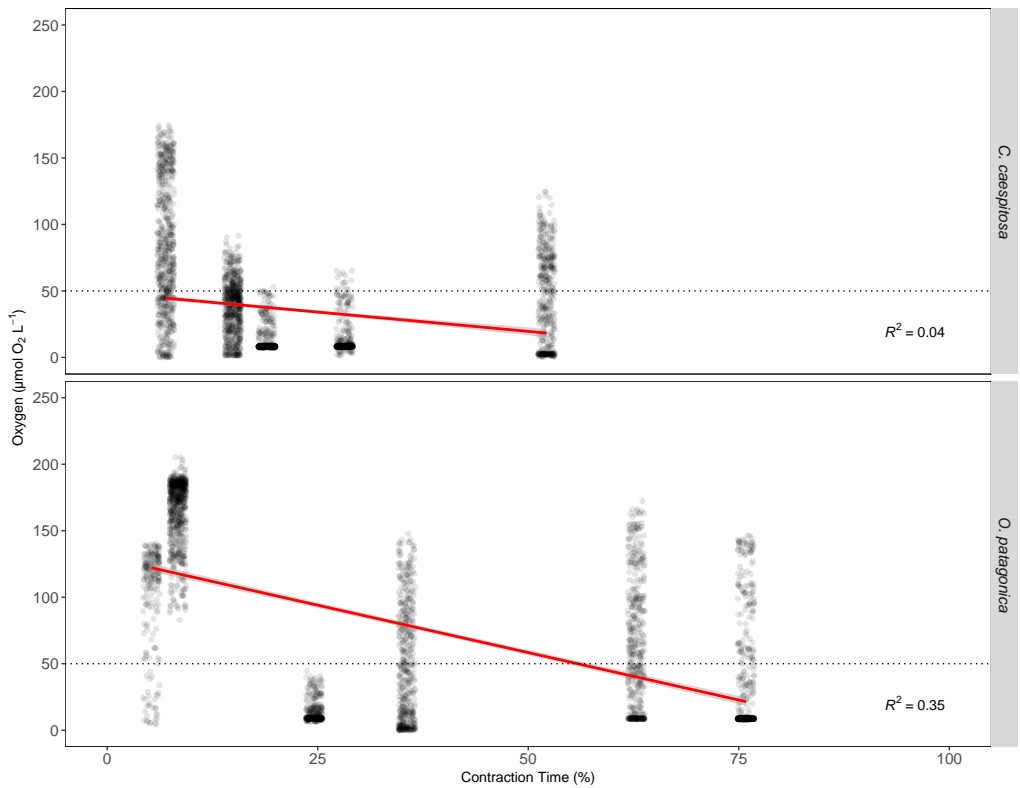

**Figure 5  The relationship between $O_2$ concentration and polyp contraction time (%) in the gastric cavity of coral specimens.** $O_2$ concentration in the gastric cavity of *C. caespitosa* and *O. patagonica* measured under dark condition relative to contraction time of the polyp. The dotted line indicates the hypoxia level at 50 $\mu$mol $O_2L^{-1}$ (equivalent to 2 mg $O_2L^{-1}$). The red continuous line indicates a linear regression model ($R^2 = 0.40$, $F = 1046$, Df $= 6257$, $p < 0.01$).

and *O. patagonica* corals resulting in a decrease of $-0.59$ $\mu$mol $O_2$ $L^{-1}$ and $-0.79$ $\mu$mol $O_2$ $L^{-1}$ per unit of contraction time, respectively, as measured locally at the position of the microsensor tip in the gastric cavity.

## Multivariate analysis

The results of the multivariate analysis (MANOVA) indicated a significant impact of Fe(III) exposure of corals on both conditions and species group ($p < 0.01$, Table S5). This influence was also evident in the PCA plots representing the data distribution of variable chlorophyll fluorescence and oxygen profile data (Fig. 6).

## DISCUSSION

While recent research has focused on the significance of iron as a limiting factor for photosynthetic endosymbionts (*Harland & Brown, 1989*; *Raven, Evans & Korb, 1999*; *Ferrier-Pagès et al., 2001*; *Rädecker et al., 2017*; *Reich et al., 2020*), knowledge of the physiological consequences on the coral holobiont remains limited. This study is one of the first investigations of the impact of Fe(III) enrichment on the photosynthesis

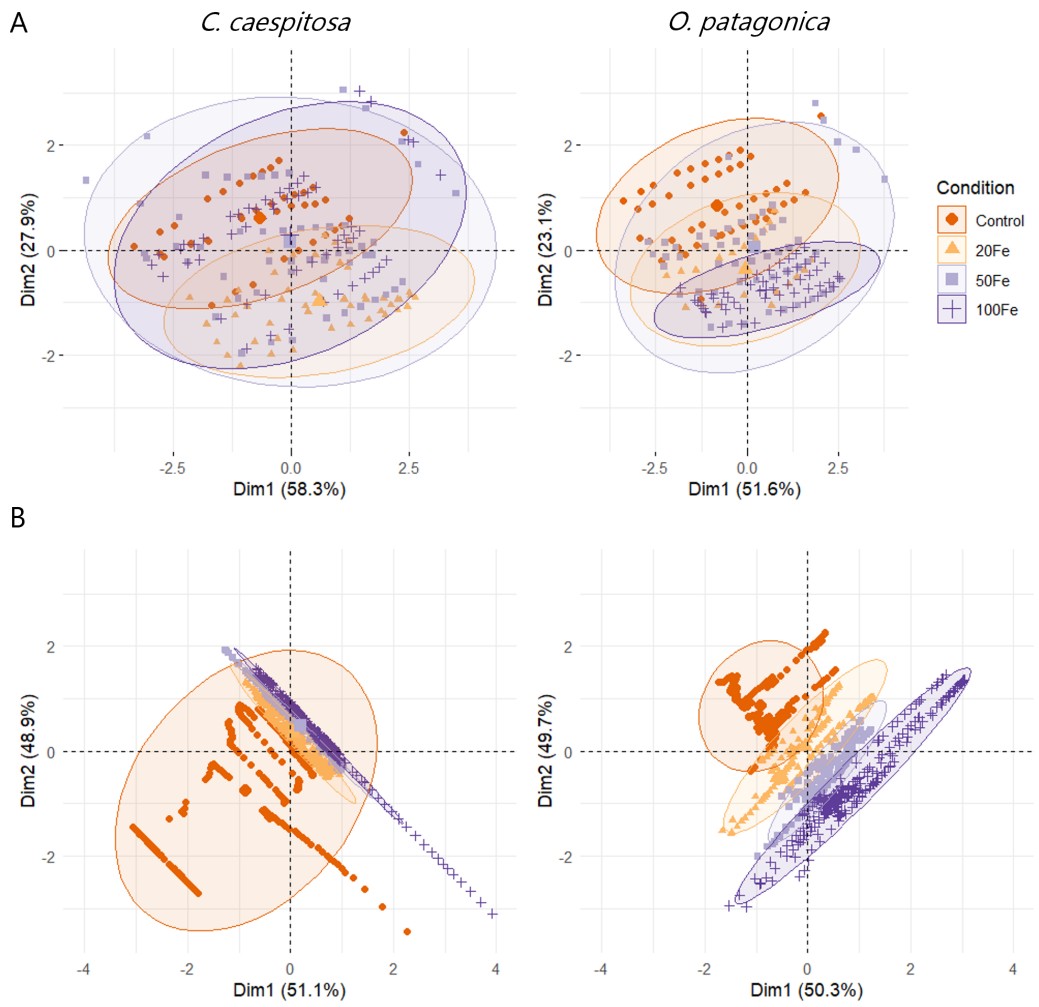

**Figure 6  Principal component analysis (PCA) plots.** (A) Chlorophyll fluorescence parameters; and (B) O$_2$ concentration in the gastric cavity of *C. caespitosa* and *O. patagonica*. Condition indicates the pulses of Fe(III) added to expose corals to 0, 20, 50, and 100 nM Fe/day.

and respiration of two Mediterranean coral species, *Cladocora caespitosa* and *Oculina patagonica*. Overall, we found that high levels (>50 nM) of Fe(III) are detrimental to the coral-Symbiodiniaceae symbiosis, decreasing the rates of photosynthesis and increasing anoxia in the gastric cavity. Lower levels (<50 nM) of Fe(III) can have beneficial effects on photosynthetic efficiency, although it is species-dependent.

The susceptibility to Fe(III)-derived enhancement of coral photosynthetic activity is species-specific. *C. caespitosa* corals inhabiting the Mediterranean Sea exhibit low frequency but high magnitude of iron records in their skeleton (*Royle et al., 2015*) suggesting that this species is naturally limited in iron but exposed to pulses of iron enrichment. Consequently, the exposure of *C. caespitosa* to increased concentration of Fe(III) (up to 50 nM, in this study) resulted in significant apparent uptake of Fe(III) (Table 1). This uptake was associated with elevated respiratory and photosynthetic activities (Fig. 2), as well

as an increase in minimum and maximum fluorescence yield of photosystem II (Fig. 3). Therefore, the exposure to low levels of Fe(III) might increase the photochemical quenching of iron-limited endosymbionts of *C. caespitosa*, leading to increased quantum efficiency and fluorescence yield (*Vassiliev' et al., 1995*; *Greene et al., 1992*). However, the overall energetic productivity (P:R) remained largely unchanged suggesting that the energy acquisition through photosynthesis was balanced by the energy consumed through respiration.

In contrast, *O. patagonica* exhibited significantly lower respiration, photosynthesis, and maximum PSII quantum yield ($F_v/F_m$) at all iron exposure levels, indicating a higher susceptibility to reduced photosynthetic performance in response to increasing Fe(III) levels. Nevertheless, increased maximum fluorescence and quantum yield of photosystem II were not linked to increased photosynthetic activity, suggesting dynamic photoinhibition of the *O. patagonica* endosymbionts under high levels of Fe(III) as a response to maintain the efficiency of energy utilization through photosynthesis (*Brown et al., 1999*; *Rodolfo-Metalpa et al., 2008*). This might indicate that *O. patagonica* is not iron-limited and increasing iron exposure may have a detrimental effect on its photosynthetic performance. Moreover, elevated iron concentrations might disrupt electron flow in the photosynthetic apparatus, as observed in the reduced rETR(II) levels in *C. caespitosa*. We speculate that these variations might be attributed to the presence of distinct Symbiodiniaceae hosted in *C. caespitosa* (*Symbiodinium microadriaticum*) and *O. patagonica* (*Breviolum psygmophylum*) (*Rodolfo-Metalpa et al., 2008*; *Casado-Amezúa et al., 2014*; *Martinez et al., 2021*; *Davies et al., 2023*). Different Symbiodiniaceae species may exhibit differences in their tolerance to iron exposure, indicating species-specific nutrient requirements (*Reich et al., 2020*; *Camp et al., 2022*). Species-specific responses to elevated Fe(III) concentration also indicate that unique biogeochemical niches play a key role in the metabolic compatibility of the coral-Symbiodiniaceae symbiosis (*Grima et al., 2022*). Distinct responses in Fe(III) uptake observed in this study might be influenced by coral evolutionary origins, and endosymbiont community structure, highlighting the importance of considering the relationship between host and symbionts traits in nutrient cycling within the holobiont.

While these findings represent a first-time observation for Mediterranean corals, elevated iron concentrations have been recognized to reduce photosynthetic capacity in tropical corals, increasing their vulnerability to coral bleaching (*Biscéré et al., 2018*). Such reductions in photosynthetic efficiency can have cascading effects on overall coral health, including growth, reproduction, and resistance to environmental stressors (*Ferrier-Pagès et al., 2001*; *Roth et al., 2021*). Elevated iron levels in seawater may thus also pose a potential stressor to Mediterranean coral holobionts.

Our measurement of $O_2$ availability within the gastric cavity of *C. caespitosa* and *O. patagonica* yielded the first insights into how respiratory activity within this compartment might be affected by enhanced dissolved iron exposure. The $O_2$ level in the gastric cavity of both *C. caespitosa* and *O. patagonica* in control conditions ranged from $\sim$100 to 300 $\mu$mol $O_2$ $L^{-1}$ (in dark and light, respectively), in contrast to previous findings of hypoxia in the gastric cavity of the tropical coral *Galaxea fascicularis* (*Agostini et al., 2012*), suggesting an enhanced capability of the investigated Mediterranean corals to exchange $O_2$ with the surrounding seawater. However, this difference might be due to the different polyp

morphology, whose length and contraction rate can affect the $O_2$ availability within the gastric cavity. A decrease in $O_2$ availability was observed when transitioning from control to Fe(III) enrichment conditions (Fig. 4), where we measured the development of hypoxic (<50 $\mu$mol $O_2$ $L^{-1}$) to anoxic conditions in both corals during daytime and overnight. These findings suggest that exposure to elevated Fe(III) reduces $O_2$ levels within the gastric cavity of Mediterranean corals, which eventually may induce hypoxia stress responses in the coral (*Alderdice et al., 2021*) and disrupt the delicate balance of the coral microbiome (*Bourne, Morrow & Webster, 2016*; *La Rivière, Garel & Bally, 2016*). Fe(III) uptake can be mediated by marine bacteria under both aerobic and anaerobic conditions (*Weber, Achenbach & Coates, 2006*; *Sandy & Butler, 2009*). The increased availability of Fe(III) in aerobic conditions might stimulate the reduction to ferrous iron, Fe(II), through bacterial siderophore complexes (*Sandy & Butler, 2009*), to be used in the redox iron cycle. Fe(II) uptake, on the other hand, is the predominant mechanism through ferrous iron uptake systems (*i.e.,* ferric uptake regulator) under anoxic conditions (*Sandy & Butler, 2009*).

Reduced $O_2$ levels in the gastric cavity might result from increased respiration of both host, microbes, and endosymbionts (*Zoccola et al., 2017*), which could inhibit the energy metabolism in coral symbiosis, reducing symbiont density and chlorophyll content, and inducing oxidative stress (*Turner et al., 2022*; *Zhang et al., 2023*), with implications for energy allocation, growth, and overall coral health (*Agostini et al., 2012*). Moreover, elevated Fe(III) concentration caused increased contraction time in the pulsation rate of the coral polyps (Fig. 5), which in turn might explain higher $O_2$ consumption within the coral gastric cavity *via* reduced $O_2$ exchange with surrounding seawater. The pulsation rate of coral polyps might play a key role in their $CO_2/O_2$ exchange and has been demonstrated to support the photosynthetic activity in zooxanthellate soft corals (*Kremien et al., 2013*). A reduction in the pulsation rate has previously been linked to pollution and eutrophication (*Loya & Rinkevich, 1980*; *Ezzat et al., 2015*; *Thobor et al., 2022*), suggesting a behavioral response of the coral host to metal toxicity.

Our observations indicate that acute exposure to Fe(III) may result in species-specific responses in corals. Our Fe exposure period was short as we wanted to mimic a pulse of dissolved Fe, in line with observations in the skeleton of *C. caespitosa* (*Royle et al., 2015*). The short-term exposure already had a beneficial effect on *C. caespitosa* and a negative effect on *O. patagonica*. It would be interesting, in a future study, to assess whether these effects on Mediterranean corals are confirmed with long-term exposure to low Fe concentrations. In addition, the Fe concentrations used for the enrichments were theoretical concentrations and the actual concentrations of bioavailable Fe(III) and its reduced form Fe(II) were most likely lower. The levels of iron to which the corals were exposed can be considered 1 to 100 times higher than typical concentrations found in coastal waters of the Mediterranean Sea, which typically receive 0.7–7 nM of dissolved iron and 0.8–14.5 nM of particulate iron (*Sarthou & Jeandel, 2001*). However, high concentrations of Fe and other heavy metals have been measured in coastal sediments of the Gulf of Genoa (*Bertolotto et al., 2005*) and the Adriatic Sea (*Orlov et al., 2022*), where corals can thrive at high density. Resuspension of sediment can occur at regular times and can release high levels of Fe into seawater. Therefore, the used iron concentrations in our study simulated acute events that might

result from sediment resuspension, exceptional runoff, or increased nutrient loads. Thus, Mediterranean corals might already be exposed to enhanced levels of dissolved iron and may not necessarily benefit from further iron enrichment. These experimental limitations should be considered when interpreting the results and underscore the need for further research with long-term exposure to comprehensively assess the effects of iron on coral ecosystems.

## CONCLUSIONS

Our study provides first insights into the intricate relationship between iron availability and coral metabolism. The impact of iron on the coral holobiont is multifaceted and contingent upon environmental factors, including ambient iron availability and the presence of other nutrients (*Harland & Brown, 1989*; *Rodriguez et al., 2016*). Considering that iron seawater chemistry is strongly influenced by environmental factors, such as ocean warming and acidification (*Shi et al., 2010*; *Shi et al., 2012*), there is a pressing need to explore how these changes may alter iron availability in the marine environment. Furthermore, the responses in different physiological parameters observed in this study highlight the complex nature of host-symbiont and host-microbiome interactions and their dependence on iron availability. Our findings emphasize the significance of considering $O_2$ availability as a crucial factor when assessing the impact of iron enrichment on coral health. Further research should focus on the exploration of the metabolic pathways affected by altered $O_2$ levels, the bioavailability of iron in the redox cycle within the gastric cavity, and the responses of photosynthetic endosymbionts as well as bacteria to iron enrichment. This will refine our understanding of the broader consequences of iron enrichment and bioavailability in coral reef ecosystems.

## ACKNOWLEDGEMENTS

We thank Saul Ciriaco, Marco Segarich, and Verdiana Vellani from the Marine Protected Area of Miramare, Trieste, and the University of Trieste (Italy) for their invaluable assistance in the sampling of *Cladocora caespitosa* in Trieste and support in transporting the colonies to the CSM laboratory. We thank Steen Færgemann Hansen from the Geoscience Department at the University of Copenhagen (Denmark) for the ICP-MS analysis. We also thank Dr. Cesar O. Pacherres from the Marine Biology Section, University of Copenhagen (Denmark) for his help with 3D printing of parts of respirometry chambers, Jonas Riff Larsen from the Niels Bohr Institute, University of Copenhagen (Denmark) for the technical support with respirometry chambers, and Elisenda Casabona Balcells for the illustrations of corals. Additionally, we express our appreciation to laboratory technicians Cecile Rottier and Maria Isabelle Marcus for their crucial contributions to the laboratory setup, and to Dr. Eric Beraud from the Ecophysiology Laboratory at the Center Scientifique de Monaco for his assistance with PAM measurements.

### Funding

This research was funded by grants from the European Union (Walter Dellisanti; Grant Agreement no. 101062810, MedCorP) and the Gordon and Betty Moore Foundation (Michael Kühl; grant no. GBMF9206; https://doi.org/10.37807/GBMF9206). The funders had no role in study design, data collection and analysis, decision to publish, or preparation of the manuscript.

### Grant Disclosures

The following grant information was disclosed by the authors:
The European Union: 101062810.
The Gordon and Betty Moore Foundation: GBMF9206.

### Competing Interests

The authors declare there are no competing interests.

### Author Contributions

- Walter Dellisanti conceived and designed the experiments, performed the experiments, analyzed the data, prepared figures and/or tables, authored or reviewed drafts of the article, and approved the final draft.
- Qingfeng Zhang performed the experiments, analyzed the data, prepared figures and/or tables, authored or reviewed drafts of the article, and approved the final draft.
- Christine Ferrier-Pagès conceived and designed the experiments, authored or reviewed drafts of the article, and approved the final draft.
- Michael Kühl conceived and designed the experiments, authored or reviewed drafts of the article, and approved the final draft.

### Field Study Permissions

The following information was supplied relating to field study approvals (i.e., approving body and any reference numbers):

Coral colonies of Cladocora caespitosa were collected from Trieste (Italy, 45.707°N, 13.712°E) using SCUBA in April 2023 (CITES IT/EX/2023/MCE/00335).

### Data Availability

The data and statistical analysis presented in this study are available at Zenodo:

- Dellisanti, W. (2024, February 23). ''Contrasting responses to increasing dissolved iron on photosynthesis and O2 availability in the gastric cavity of two Mediterranean corals.'' Time-lapse videos. Zenodo. https://doi.org/10.5281/zenodo.10698045

- Dellisanti, W. (2023). ''Contrasting responses to increasing dissolved iron on photosynthesis and O2 availability in the gastric cavity of two Mediterranean corals.'' Dataset [Data set]. Zenodo. https://doi.org/10.5281/zenodo.10697973

- Dellisanti, W. (2023). "Contrasting responses to increasing dissolved iron on photosynthesis and O2 availability in the gastric cavity of two Mediterranean corals." R code for statistical analysis. Zenodo. https://doi.org/10.5281/zenodo.10697995

**Supplemental Information**

Supplemental information for this article can be found online at http://dx.doi.org/10.7717/peerj.17259#supplemental-information.

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
