# Peer review of "Contrasting effects of increasing dissolved iron on photosynthesis and O2 availability in the gastric cavity of two Mediterranean corals"

_PeerJ, doi:10.7717/peerj.17259_

## Round 0.1 · original submission · Major Revisions

Three recognized experts have evaluated your manuscript and their comments can be seen below. All agree that this is a well written and structured paper with a a very good experimental design. However, a number of issues have been pointed out and while they are mostly minor, I have chosen major revision simply due to the fact there are many observations that need to be considered. Please ensure that you upload the corrected (tracked changes) version of the manuscript as well as a clean version and a detailed rebuttal letter.

·

Basic reporting

Increasing dissolved iron limits photosynthesis and reduces the O2 availability in the gastric cavity of two Mediterranean corals by Walter Dellisanti.
Dellisanti and colleagues explore for the first time in this ms the effect of Fe(III) seawater enrichment on the photosynthetic performances of two most relevant scleractinian corals in the Med Sea. Using respirometric chambers and microsensors into the oral cavity, the Authors showed a species-specific effect of high concentrations in Fe leading to changes in the photosynthetic parameters and polyp contraction. Differences between the two coral species were used to infer their potential Fe limitation in the nature limiting their photosynthetic performances.
This study is well written, data are clearly presented with nice figures and nothing to say also about the statistical approach used. I think this paper is ready to be published and I only have some details I invite the Authors to consider as minor changes.
Testing the effect of increasing concentrations of metal, pollutants etc in aquaria is a valid approach to push the organism toward a shock response and better explore the mechanism involved. I do not have a particular criticism and my two major concerns about the Fe-form bioavailable for the coral (which is always difficult to assess as it changes sometimes at unknown rates), and its concentration, were honestly presented by the Authors as potential limitations of this study at the end of the discussion. Since sometimes people are lazy and only read the abstract of studies, maybe it would be better to give an appreciation of these limitations (at least the concentration of the tested Fe) from the beginning.
The discussion could be gained if the different responses of the two species, and their limitation in Fe, were discussed in terms of their actual distribution, abundance and maybe geographical expansion. I know it is a bit difficult and not the scope of this study. This is just an idea because I worked in the past with these two species and I remember that they can live quite close to rivers, especially on coastal intertidal, harbours etc. Maybe, it would be possible to find in the literature one, or two concentrations of coastal Fe where one of these species is particularly abundant… It could reinforce the mechanistic effect measured in aquaria with unrealistic concentrations and link results to reality. I repeat it is just an idea.
Some details need to be fixed. Maybe some of the details I did not find were already in the text but I did not pay enough attention. Please find what I found below.
L 92-93. The introduction is very well written. Here, the Authors mentioned the concentrations of Fe in the Med and reported the concentration measured at one deep hydrothermal vent, which was at a value used in the present experiment. I found this example a bit extreme. I do not think it is necessary and honestly irrelevant to justify the concentration used. Both species are shallow-water species (although Oculina seems to expand deeper, a little bit at least). As I suggested, maybe find extreme values due to runoff and river input (La Spezia Magra River as an instance?).
L 116. I am aware that Oculina has spread a lot during the last decade but I ignored that it arrived in La Spezia. Do you have any references for this new entry?
L130, method in general. I would appreciate some more details to better understand how the Fe solution was injected and maintained in the experimental tanks. Any seawater renewal during the 1-w incubation? At what rate? Did the Authors use a peristaltic pump for that? How the uptake rate was measured? When?
L162. Please, could you specify how the Pg:R was calculated? Indeed, it could be calculated simply by considering the Pg and R measured in the light and the dark or calculated by considering the 12h light (Pg) and 24h dark (R). This might eventually change the final data but not the main finding.
L193. Please, could you give more details of the custom-made flow chamber and in particular how a laminar water flow was obtained?

Experimental design

All in Basic reporting

Validity of the findings

All in Basic reporting

Additional comments

All in Basic reporting

·

Basic reporting

The manuscript is well written, referenced and well structured. The figures are fine. The hypothesis are well defined and tested.

Experimental design

The manuscript by Dellisanti et al. report on the effect of iron on the corals C caespitosa and O. patagonica. This study is highly original for three main points: 1) there are few studies on the effect of irons on corals, 2) mediterrannean corals are kind of understudied, 3) the authors in addition to "classical" metabolism measurement, included measurements of oxygen levels in the gastric cavitiy and linked them to behavioral changes, I found this very original and interesting. Overall the paper is well written and easy to read.
Some criticism could be done on the experimental desgin where only one tank per treatment is used but it is clearly written and therefore it is not a blocking problem in my opinion. However I have another point that I suggest should be investigated and clarified before publication regarding the stat analysis of the oxygen levels in he gastrovascular cavity.

Validity of the findings

The discussion and conclusion are based on the findings, no problem on that point. There is some limitation on the replication but it is acknowedged. Maybe the stat could be better. I provide some suggestion in the detailed comment below.

Additional comments

## Materials and Methods

line 114: please clarify the number of colonies that were sampled.

Figure 1: It is difficult to see and understand the skeleton structure in the diagram for C. caespitosa. Specifically, I don't understand if the polyp wall have some skeleton in it (like Galaxea fascicularis for example) or not (like Goniopora species for an extreme example).

line 126: four replicated tank. Maybe remove replicated, as if I understood well there is only one tank per treatment so there are not replicate tank statistically speaking. Many would say that the experimental unit should be the tank (Hurlbert, 1984) however many physiologist use a smilar design as yours and are not criticized for it, moreover Grotoli et al. in (2021) do state that that for accute experiment a minimum of two tanks per treatment should be used... to avoid criticism and make the limitation clear I wold suggest to clarify the tank volume turnover and the number of ramets per tank.

line 159: add a space before the parenthesis "technique (Marsh, 1970)" but more importantly: "[...] and the area was calculated [...]": do you mean the area of the aluminium? Interesting way, in the original methods and what I usually do (I assumed everyone did the same) the aluminium is weighted and the weight is converted to surface. I don't see a problem in using imagej (as long as the piece of aluminium were flattened) but as it seems unusual, for clarity I would add "the surface area of the pieces of aluminium that were used to cover living tissue was determined with the ImageJ software.

## Results

line 234 and others: consider adding stats info, test type, chi-sqaure and df. It is not 100% required as it is in the table but I personally like to be able to see it directly.

line 280: Regarding comparisons of oxygen concentration along the depth of the GVC: How was that tested, did you use the mean across the length of the GVC or the mean of the lowest point of the profile? This should be clarified in the method too. Also need to refere to the different panel, so here Fig4-A

line 288: "how was that tested, did you use the mean across the length of the GVC or the mean of the lowest point of the profile? This should be clarified in the method too." Need to refer to Fig4-B. Also could you clarify if this data set was taken under light or both light and dark? If both light and dark, please add to the figure the different period of illumination.

line 291: the availability of O2: is that the overall mean? so the mean of the time series, and mean of the different colonies. In that case it would be preferable to use a linear mixed model (or generalized linear mixed model) as the oxygen concentration within the same polyp measurement are repeated measurement. You would need to add the polyp as a random factor, I would suggest on the intercept. In R language: lme(Oxygen ~ treatment + 1|polyp)

line 300: "p < 0.01, Fig. 5, Table S3": here again I think you need to include the polyp as random factor I assume. Just looking at the data, I think it would make the results even clearer. It is well possible that the microelectrode was not inserted at the same depth for all measurement and therefore you end up with different levels of oxygen, but their dynamic should still be dictated by the contraction of the polyp (aka ventilation).

## Discussion

line 365: More of a note that a comment, but maybe worth adding something on it: The difference observed may be due to different morphology. The polyps of G. fascicularis are calcified until the oral disk, and it is mostly impossible for them to completely retract. Therefore G. fasciularis may be not typical with a low ventilation rate of GVC. From Fig.1 I am not sure if C. caespitosa is calcified until the oral disc or not, nevertheless eyeballing the data it seems that C. caespitosa show generally lower concentration in oxygen than O patagonica, and they also have longer polyps, which once again highlight the importance of the polyp morphology in determining the amount of ventilation in the GVC. Up to you to add or not.

## References

The references need to be checked. In many instances the doi point to figures or supplementry. For example in Agostini et al., the doi is 10.1007/S00338-011-0831-6/FIGURES/6. instead of 10.1007/S00338-011-0831-6. Also it seems there is a problem with Annual Review reference format, for example: "Behrenfeld MJ, Milligan AJ. 2013. Photophysiological Expressions of Iron Stress in Phytoplankton. https://doi.org/10.1146/annurev-marine-121211-172356 5:217–246. DOI: 10.1146/ANNUREV-MARINE-121211-172356."

Reviewer 3 ·

Basic reporting

[a] intro/background

Lines 90-100: the first half of the intro focused on tropical reef-building corals (which is timely and suitable) but the dissolved iron concentrations in the Mediterranean are considerably higher than the Caribbean & Indo-Pacific. This needs to be addressed in the intro/discussion to better contextualize the [Fe] selected.

See references within:
Ocean Data View https://odv.awi.de/data/ocean/; eGeotraces https://egeotraces.org/?group=Dissolved%20Trace%20Elements,variable=Fe%20dissolved

Overall, introduction was well written and does a thorough job introducing iron cycling in coral symbiosis as well as the Mediterranean temperate corals used for the study.

[b] figure quality, labelling
Add Fe concentration to figure labels.

Figures 2-6 appear pixelated in the PDF.

Figure 4B: indicate the light/dark period in the caption

The [Fe] of tanks should be the first component of the results section and needs to be presented as either a data table or figure in the main document (alluded to in methods L137-141). Reporting the [Fe] of the control seawater treatment in particular is crucial to verifying the integrity of the experimental design.

[c] raw data is not available

Experimental design

[a] research gap

This study addresses an exciting dearth in research in the temperate coral ecophysiology and trace metal biogeochemistry realms of marine science. I enjoyed reading the paper and think it will be of broad interest to the peerJ readership.

[b] methods appropriate

See major concerns about inclusion of control/tank Fe concentrations in the results. The methods the iron uptake rates (results presented in table 1) are missing from the main text.

Validity of the findings

Underlying data needed before validity of findings can be fully assessed.

Additional comments

Interspecific variation of corals to iron limitation is a major finding of the study – it is well-communicated in the abstract, results, and discussion but disjointed from the title

See Grima et al 2022 Coral reefs – I think the incorporating concepts on biogeochemical niches (BN) and how they pertain to coral trace metal needs will help strengthen the discussion of intraspecific variation https://link.springer.com/article/10.1007/s00338-022-02259-2

Discussion L349-352 – omit clade/phylotype Symbiodiniaceae nomenclature.
See Davies et al 2023 https://peerj.com/articles/15023/ for best practices for incorporating Symbiodiniaceae nomenclature into writing. See LaJeunesse et al 2022 (http://dx.doi.org/10.1080/09670262.2021.1914863) for more on the diversity of temperate Symbiodiniaceae in the Mediterranean.

---

## Round 0.2 · accepted · Accept

Thank you for taking the time to answer all the concerns and suggestions so thoroughly. I am satisfied with the changes you have made to the manuscript and propose that it be accepted for publication.

Reviewer 3 ·

Basic reporting

I thank the authors for their efforts revising the ms. I have no further comments

Experimental design

n/a

Validity of the findings

n/a